# Position: Model identity in machine learning is a convention, not a property

**Vacslav Glukhov** [* 1]

## Abstract

Treating the outcome of machine learning as a stable, identifiable artifact is implicit in language, tooling, and governance. This position paper examines whether a trained system admits context-appropriate criteria of identity. We show that neither functional behavior nor internal structure suffices: behavioral equivalence is underdetermined by finite data, while modern architectures admit multiple, structurally distinct realizations of the same function. Consequently, practices that treat learned systems as stable objects presuppose equivalence relations that are rarely made explicit. I do not propose abandoning such practices. Instead, I articulate the reasonably minimal conditions under which identity claims grounded in behavior, structure, or training process can be meaningfully interpreted, with implications for reproducibility, traceability, and governance.

## 1. What do we normally assume machine learning produces?

Current learning practice relies on explicit procedures as much as on implicit assumptions. These assumptions are rarely stated as principles, yet they shape how learned systems are trained, evaluated, deployed, and governed.

Machine learning is commonly understood as the acquisition of knowledge in a narrow sense: a parameterized function is constructed to represent aspects of real-world phenomena, and the state of knowledge is assessed through an empirical error of representation. Improvements in learning are therefore identified with reductions in this error.

The learning process is further assumed to converge, which is understood as a state in which the acquired knowledge cannot be substantially improved by modifying the function. While convergence is typically measured by a loss function that reflects empirical error, it is assumed to occur in parameter space. The existence of formal convergence guarantees is treated as a desirable property of a learning method; in their absence, decreasing representation error serves as a practical surrogate.

Once trained, a learned system is deployed as a parameterized function embedded in a larger software system. Continuous integration and deployment practices assume the existence of procedures for change control, testing, and rollback. These procedures presuppose that the learned system exists as a deployable artifact whose modifications can be detected and evaluated.

To maintain referential integrity between a learned system and its surrounding components, the parameterized function is named and versioned. Upstream and downstream processes rely on this identity to coordinate data flow, testing, and validation. The practice of continuous deployment further requires that changes to the learned system be identifiable as updates, upgrades, or replacements in order to trigger appropriate testing and release mechanisms.

Learned systems are also assumed to exhibit stability over time that reflects the stability of their target real-world phenomena. Gradual changes in the underlying phenomena are expected to result in gradual changes in acquired knowledge. Retraining is initiated when representational quality falls below an acceptable threshold, with the implicit assumption that the system being retrained remains the same system – updated rather than replaced.

Finally, learning practice assumes a degree of reproducibility and replicability. Under identical learning conditions, a system is expected to acquire the same knowledge. Under changed conditions, we expect the system to replicate essential aspects of previously acquired knowledge. These expectations support comparison across training runs and across time.

From the preceding, it is evident that treating learned systems as stable artifacts is embedded in the language, tooling, and governance of learning practice. It enables coordination across complex technical and organizational environments. This assumption is not confined to engineering practice; it is increasingly formalized in governance frameworks, including regulatory regimes that require explicit identification

[1]ItoFlow, London, UK. Correspondence to: Vacslav Glukhov <vacslav@itoflow.ai>.

*Proceedings of the 43$^{rd}$ International Conference on Machine Learning*, Seoul, South Korea. PMLR 306, 2026. Copyright 2026 by the author(s).

and traceability of AI systems as deployable entities.

Treating the outcome of learning as an object presupposes a form of identifiability, but not necessarily uniqueness in an absolute sense. Objects are individuated relative to a context: by the distinctions that matter, by the operations that can be performed, and by the criteria under which sameness and difference are defined. In some contexts, objects are interchangeable; in others, fine-grained distinctions are essential. Applied to learning systems, this leads to a question: do trained systems admit stable, context-appropriate criteria by which they can be compared, distinguished, referenced, and acted upon?

In other words, is treating a learning outcome as an object justified? Answering this requires inspecting whether a trained system admits a well-defined identity in the first place. We will now examine this question from two complementary perspectives: functional and structural.

## 2. Can a learned system be identified by its function?

One may object that no such identity is actually required. From a pragmatic standpoint, it may be sufficient to define a learned system purely at the level of input–output behavior and to attach governance to the processes that produce and validate that behavior. On this view, retraining, replacement, and redeployment are unproblematic so long as externally observable performance remains acceptable. The internal identity of the learned system is then a matter of convenience rather than principle: if two trained systems implement the same input–output mapping, one may regard them as the same for all practical purposes. This view underlies many pragmatic approaches to deployment and governance, where internal structure is treated as irrelevant so long as externally observable performance characteristics remain acceptable. On this account, objecthood is relaxed to functional equivalence.

While this perspective is coherent and often regarded as sufficient in practice, it entirely depends on whether functional equivalence can serve as a reliable, operationally meaningful criterion for learning systems.

### 2.1. Limits of functional equivalence in data-centric learning

Functional equivalence is a demanding requirement for data-centric learning systems.

Learned models are constrained to a finite, typically sparse subset of their input space, defined by available data and evaluation procedures. However, agreement between systems on a set of observed inputs does not entail agreement elsewhere. Two systems interrogated on the same data set,

no matter its size, and exhibiting the same input-output mapping, can still differ on inputs not yet examined, a property assured by the universal approximation theorem.

As a result, there is generally no operational procedure for determining whether two trained systems implement the same input–output mapping beyond the data points explicitly tested. While the pointwise testing can falsify equivalence, it cannot establish it, a property necessary for governance of third-party learned systems in high-security, high-stakes, and regulated environments. If guarantees of equivalence are required, behavioral testing alone, however massive, is insufficient. Functional identity is therefore undetermined, not merely difficult to empirically verify.

### 2.2. Contrast with traditional software systems

The appeal of functional equivalence is reinforced by analogy with conventional software. In standard software systems, the presence, absence, and expected behavior of a function can and often must be specified and tested exhaustively. While complete verification is not always practical, the system presents itself as, at least in principle, a fully testable artifact whose behavior is fixed by its specification.

Learned systems differ in kind. Their behavior is not explicitly specified but inferred from data, and correctness is assessed statistically rather than exhaustively. As a result, the conditions that make functional equivalence a viable criterion of identity in traditional software systems are not generally available in data-centric learning.

Since functional equivalence generally cannot be established from the evidence available to data-centric learning systems, functional behavior does not provide stable criteria of identity.

But if its function cannot identify the learned system, can its identity be recovered by examining its internal structure? The next section considers this possibility.

## 3. Can a learned system be identified by its internal structure?

It is plausible that identity can be recovered at the level of parameters, representations, or internal organization. This expectation is natural and underlies many standard practices in model governance. The analysis in this section shows that this expectation is misplaced. In modern learning architectures, internal structure does not uniquely determine learning outcomes. Instead, the structure introduces additional degrees of non-identifiability.

In a narrow sense, a machine learning system acquires knowledge by constructing a parameterized function. This function is defined by its architecture and its parameters.

**Architectures** are typically specified explicitly and admit clear criteria of equivalence. Modulo trivial transformations – such as algebraic rearrangements or equivalent code realizations – an architecture possesses a stable identity. Changes in architecture can be detected, tracked, named, and versioned by standard means. In this limited sense, the model's architecture behaves as an object.

**Parameters** are a different matter. The mapping from parameters to induced behavior is generally many-to-one, even in classical parametric models. For example, in principal component analysis, the learned representation is invariant under sign changes of individual components, leading to many distinct parameterizations that encode the same solution. Here, as in more complex learning systems, internal descriptions outnumber the behaviors they induce.

Modern architectures multiply such redundancies. They are built from collections of relatively simple and interchangeable computational elements, arranged in layers, blocks, or stacks. The operations performed by these elements are chosen for computational efficiency and scalability and rely heavily on linear transformations and pointwise nonlinearities. This design choice introduces structured invariances: transformations of internal elements that leave the overall input–output behavior unchanged.

One such invariance arises from **permutation symmetries**. In a two-layer fully connected network, permuting the hidden units within a layer – together with the corresponding permutation of incoming and outgoing weights – leaves the represented function unchanged. The same phenomenon extends to deeper networks, where permutations can be applied independently within multiple layers. The number of such functionally equivalent realizations grows combinatorially with network width and depth, making non-uniqueness not an edge case but a structural property of modern architectures.

If discrete symmetries already eliminate uniqueness, continuous symmetries remove any privileged internal coordinate system. Continuous orthogonal transformations – such as **rotations of vector embedding spaces** that preserve a dot product – generate infinitely many distinct internal representations that implement the same function. Any architectural component that relies on such inner products inherits this invariance under joint transformations.

**Normalization mechanisms** further amplify this. For example, layer normalization explicitly removes scale information and is invariant under uniform shifts. As a result, entire families of internal states collapse to the same functional behavior, while remaining distinct in parameter space.

Taken together, these considerations show that internal structure does not provide reliable criteria for individuation in learned systems. Discrete symmetries undermine uniqueness by allowing multiple distinct parameterizations to realize the same behavior, while continuous symmetries eliminate any privileged internal coordinate system and generate uncountably many functionally equivalent internal states.

Thus, internal descriptions outnumber the behaviors they induce, and identity claims based on structure implicitly select a single representative from the class of functionally equivalent systems. Inspection of the internal structure does not recover an underlying object. Rather, it exposes the equivalence relations under which such an object would have to be defined.

## 4. What now?

Reiterating the demands of practice, treating the outcome of learning – a learned system – as an object is presumed not only possible, but necessary for its governance and reliable deployment in complex technical and organizational environments. Modern regulatory regimes explicitly require "identification and traceability of the AI system" (EU AI Act, Annex VIII), making some notion of system identity a legal requirement.

The arguments in Sections 2 and 3 do not suggest that model identification, tracking, and comparison are futile. They do imply that they are not free. Any claim that treats a learned system as an object – whether at the level of representations or across deployments – presupposes criteria of individuation that are rarely, if ever, stated. An appropriate response is not to ridicule or ban such claims, but to make their preconditions explicit.

What follows is a set of reasonably minimal commitments: if one wishes to speak about internal representations or compare models across retraining as stable properties, one must first specify the equivalence relation under which such comparisons are intended to hold, and the level of granularity (functional, structural, or procedural) at which identity is asserted.

If **functional equivalence** (or lack thereof) is taken as the basis for identity, one must specify the functional bands, partitions, or regions of the input–output space that make alignment between two or more learned systems logically possible, regardless of their internal structure.

If **internal structure** is claimed to be sufficient for identification, one must supply a gauge fix—that is, an explicit choice of equivalence that selects a representative within the class of functionally equivalent parameterizations. This need not yield a unique canonical form; it specifies which distinctions are treated as irrelevant in a given context.

Finally, if the **learning process** itself is claimed to provide the required identification, then process governance must supply the corresponding guarantees: the criteria under

which two training runs are to be regarded as producing the same system, and the conditions under which deviations are to be detected and justified.

In practice, identity is rarely binary. Systems are often treated as "updates" or "new models" depending on the level of equivalence invoked: a retrained or quantized model may be considered the same under behavioral tolerances, yet different under structural criteria. This suggests a hierarchy of equivalence relations rather than a single notion of identity.

## 5. Conclusion

This paper has argued that the outcome of learning does not, in general, admit objecthood in the sense presupposed by much of current practice. Neither behavioral performance nor internal structure provides stable, context-appropriate criteria of identity, and attempts to recover such criteria encounter structural non-identifiability rather than merely practical limitations.

The proper response is not to abandon model comparison or governance, but to recognize that these activities rely on choices of equivalence that are rarely made explicit. Once objecthood is no longer assumed, the burden shifts from objectifying "the" learned system to specifying the conditions under which different realizations are to be regarded as the same or different *for a given purpose*.

The central claim is therefore not a rejection of existing practice, but a constraint on its interpretation: meaningful claims about learned systems require that their criteria of identity be stated rather than presumed.

## 6. Call to Action

The position advanced in this paper is not a proposal for a new modeling paradigm, but a constraint on how claims about learned systems should be interpreted. If neither behavior nor internal structure supplies intrinsic identity, then any practice that treats a trained system as an object must make its criteria of sameness explicit. The following actions are therefore not prescriptions of method, but minimal obligations for clarity, reproducibility, and governance.

### For researchers and authors

When making claims that compare models across training runs, datasets, or deployments, authors should state the equivalence relation under which such comparisons are intended to hold. Whether identity is grounded in behavioral tolerance, internal structure, or training procedure, the relevant notion of sameness should be specified rather than left implicit. This applies equally to claims about reproducibility, transfer, robustness, and model updates, and is also relevant to claims of a model's superior or inferior performance, which should specify tolerances under which two models' performance is considered the same.

### For benchmarks and evaluation frameworks

Evaluation protocols should distinguish between falsifying and establishing equivalence. Where claims of functional identity are implied, benchmarks should clarify the regions of the input–output space over which equivalence is assessed and the tolerances under which models are regarded as "the same." Absent such specification, functional performance similarity should be interpreted as task adequacy, not identity.

### For tooling and infrastructure

Model versioning systems, registries, and deployment pipelines should make explicit the criteria under which one model instance is considered an update of another rather than a replacement. When identity is defined by internal structure, any gauge-fixing conventions – such as parameter normalization, selective layer pruning, alignment, or canonicalization of embedding bases – should be documented as part of the model artifact. When identity is process-based, the governance rules that define equivalence between training runs should be made explicit.

### For governance and regulation

Where traceability, accountability, or certification depend on identifying "the" AI system, regulatory and organizational frameworks should specify the basis of that identification: behavioral tolerances, structural conventions, or procedural guarantees. Treating identity as self-evident obscures the conditions under which compliance is assessed and undermines the interpretability of audit and oversight mechanisms.

Taken together, these actions do not resolve the structural non-identifiability of learned systems. Their purpose is not to re-establish objecthood, but to ensure that claims which rely on identity are stated in terms that make their underlying assumptions explicit.

## 7. Alternative Views

The position advanced in this paper challenges a set of assumptions that are deeply embedded in both research practice and deployment. Several alternative views reject or bypass the need for object-level identity in learned systems, each in a principled and often pragmatically successful way. This section considers three such views and clarifies why, while coherent within their own frames, they do not undermine the structural claim made here.

## 7.1. Functional equivalence is sufficient

A common view holds that a learned system needs only be identified by its externally observable behavior. If two systems implement the same input–output mapping to within acceptable tolerances, they may be treated as the same for all practical purposes. This position underlies many deployment and governance practices in which internal structure is regarded as irrelevant so long as performance metrics are met and operational processes are standardized.

This view is attractive because it is operational and task-oriented. It aligns with engineering practice, supports modular replacement, and avoids commitments to internal representations. However, as argued in Section 2, functional equivalence in data-centric learning is not a stable or verifiable criterion of identity. Agreement on observed inputs does not entail agreement beyond tested regions, and there is no general procedure for establishing equivalence of input–output mappings in high-dimensional or structured domains. A modification of a learned system whereby it passes the countable set of tests but produces a novel behavior, e.g., adversarial, in response to a particular input, is a concern in high-security, high-stakes, and regulated systems. As a result, behavior can at best falsify identity, not establish it. The functional view therefore does not provide the context-independent criteria of sameness required for treating learned systems as objects; it presupposes an identity that it cannot itself secure.

## 7.2. Internal structure defines the model

A second view locates identity in the internal organization of a trained system. On this account, parameters, representations, or architectural structure constitute the learned model, and sufficient inspection of these elements should in principle recover its identity. This perspective underlies many practices in model comparison, debugging, and analysis of learned representations.

This view is seemingly compelling because internal structure is concrete, inspectable, and naturally aligned with how models are implemented and stored. Yet, as shown in Section 3, modern learning architectures exhibit pervasive non-identifiability: discrete symmetries yield multiple parameterizations corresponding to the same function, and continuous symmetries eliminate any privileged internal coordinate system. Normalization mechanisms further collapse large families of internal states to identical behavior. Internal descriptions therefore outnumber the behaviors they induce, and there is no canonical representative within these equivalence classes. Structural inspection does not recover a unique object; it exposes additional degrees of freedom. Consequently, internal structure cannot ground identity in the sense required for objecthood.

## 7.3. Process-level governance is enough

A third view holds that identity need not be defined at the level of behavior or structure at all. Instead, governance can be attached to the learning process itself: training pipelines, evaluation protocols, documentation standards, and audit trails. On this view, reproducibility, traceability, and accountability are properties of procedures rather than of models as objects. If the process is well specified and controlled, the identity of any particular trained instance is of secondary importance.

This perspective is both practically successful and increasingly institutionalized. It supports large-scale experimentation, continuous integration, and regulatory oversight without requiring a canonical model representation. However, it does not eliminate the underlying issue of identity. Instead, it relocates it. Process-based governance still requires criteria for when two runs are to be regarded as producing "the same" system, for when deviations constitute updates rather than replacements, and for how outputs are to be compared across time and context. These judgments necessarily invoke equivalence relations, even if only implicitly. Process alone cannot supply identity unless its own criteria of sameness are made explicit.

## 8. Notes

**On parameter-space redundancy and non-uniqueness.**

(Sagun et al., 2017): The paper analyzes the Hessian spectrum in over-parameterized neural networks and documents large flat components in the loss geometry, offering a concrete technical illustration of the many-to-one relationship between parameter configurations and induced behavior. See 1706.04454.

**On representational non-uniqueness across training runs.**

(Morcos et al., 2018): The authors develop a representational similarity analysis based on canonical correlation and show that internal representations can vary substantially across networks trained on the same task, despite comparable performance. See 1806.05759.

**On identifiability as a formal notion.**

(Allman et al., 2009): The paper treats identifiability in latent structure models and clarifies when parameters can and cannot be uniquely recovered from observable data; their framing situates non-identifiability as a structural property of model classes rather than an implementation accident. See JSTOR+1.

**On reproducibility and process-oriented controls.**

(Pineau et al., 2021): The authors summarize the NeurIPS

reproducibility program and its findings, providing an institutional reference point for the fact that modern ML practice often relies on procedural controls and reporting standards rather than on intrinsic criteria of model identity. See jmlr.org+1.

**On collapsing symmetries in neural network parameter spaces.**

(Sorensen, 2020): Thesis directly engages the problem of symmetries in parameter space and the idea of collapsing or quotienting them, closely aligned with the concerns about non-identifiability emphasized in Section 3. The author proposes a pragmatic and efficient heuristic for gauge fixing in permutatively invariant systems. See dash.harvard.edu+1.

(Glukhov, 2024): In sections 5 and 6, I provide a more formal treatment of non-identifiability, symmetry, and equivalence classes in deep learning models. I also discuss practical gauge fixes for a class of symmetries. See 2411.07008.

**On regulatory demands for traceability.**

The EU Artificial Intelligence Act (European Union, 2024), and its Annex VIII explicitly require "identification and traceability of the AI system," motivating the practical pressure toward object-like identity even when such identity is structurally underdetermined. See EUR-Lex+1.

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
