# OpenReview forum: "Position: Model identity in machine learning is a convention, not a property"
_ICML.cc/2026/Position_Paper_Track — ICML 2026 Position Paper Track regular_

### Official Review · Reviewer_KDb6 · 2026-02-15

**Significance:** 3
**Argument Clarity:** 2
**Rating:** 2
**Confidence:** 3

**Questions:**

Please see the Weaknesses part.

**Alternative Views Section:**

Yes

**Compliance With Llm Reviewing Policy A Conservative:**

Affirmed.

**Discussion Potential:**

3

**Paper Summary:**

This position paper argues that model identity in modern machine learning is a convention rather than an inherent property. It explains that behavior alone cannot reliably certify sameness from finite evaluation, and internal structure is also unreliable because different parameterizations can implement the same behavior. Based on this, it advocates defining model identity through explicit, context-appropriate criteria so that reproducibility, auditing, versioning, and governance can be handled consistently.

**Position:**

Yes

**Position In Title:**

Yes

**Related Work:**

2

**Strengths And Weaknesses:**

**Strengths**

1. The paper raises a discussion about how the community talks about trained models as stable artifacts and what is actually justified by evidence, tooling, and governance practice.
2. The call to action is practical. It does not ask the community to stop versioning or comparing models, but to state the equivalence relation being assumed and to document conventions when identity is claimed through structure or process.


**Weaknesses**

1. This paper is mostly conceptual, and it provides little concrete evidence or formalization beyond high-level reasoning and discussions. The overall content volume of this paper appears rather thin.
2. The gauge-fixing idea is somewhat under-specified as an engineering proposal. The paper does not spell out what canonicalization would look like for common architectures, what guarantees it would provide, or when it is feasible versus ill-posed.

**Support:**

2

---

> ### Author Rebuttal · Authors · 2026-03-26
>
> We thank the reviewer for engaging with the core question and for highlighting concerns about concreteness and engineering relevance.
>
> **On “thinness” and lack of formalization.**
>
> This submission is explicitly framed as a _position paper_. Its contribution is not a new formalism or algorithm, but a structural claim: treating learned systems as stable objects presupposes an equivalence relation that is rarely made explicit. The goal is to constrain how identity claims are interpreted, rather than to replace existing practices with a new technical framework.
>
> We agree that the notion of “minimal obligation” can be clarified. Our intent is not to require a complete canonicalization of model representations. Rather, the requirement is that the basis of equivalence be made explicit at the level at which identity is asserted. Concretely:
>
> If identity is grounded in behavior, the relevant input regions and tolerances should be specified.
> If in structure, the invariances being ignored (e.g., permutations, rotations) should be documented, even if not fully resolved.
> If in process, the criteria under which two runs are considered equivalent should be stated.
>
> In this sense, a “gauge fix” does not require a unique canonical representation; it defines the equivalence class within which identity claims are made.
>
> **On feasibility and scalability.**
>
> We agree that fully resolving symmetries is generally infeasible for modern architectures. Our argument does not depend on such feasibility. The presence of large equivalence classes is precisely why identity cannot be treated as intrinsic. The practical implication is therefore not to compute canonical representatives, but to make explicit which distinctions are being ignored in a given context.
>
> **On practical grounding.**
>
> In current deployment pipelines, identity is often defined at the artifact level (e.g., via hashes of model weights), ensuring referential integrity. This notion is operationally effective but strictly finer than functional or structural identity: different artifacts may implement the same function or belong to the same symmetry class. When identity claims are transferred across contexts (e.g., retraining, compression, or auditing), mismatches between these notions can arise unless the underlying equivalence is specified.
>
> **On “update” vs “new model”.**
>
> A related issue concerns when a system is treated as an “update” rather than a new model. In practice, this distinction is operational but implicit. In our framing, it corresponds to a hierarchy of equivalence relations: a system may be considered the same under a coarse behavioral tolerance but different under a finer structural criterion. For example, pruning or quantization may preserve task-level behavior while altering parameters, supporting different identity judgments depending on context. Making these criteria explicit is part of the proposed obligation.
>
> We will revise the manuscript to clarify that the proposal is not to impose a universal canonicalization, but to make explicit the context-dependent equivalence relations already underlying current practice.

---

> > ### Author Rebuttal · Reviewer_KDb6 · 2026-04-02
> >
> > Thanks for the authors' rebuttal.
> > "Position paper" is not a justification for thin content. It is recommended that the author review the "position paper" from previous years of this conference for the expected level of content richness.

---

### Official Review · Reviewer_JeLy · 2026-03-12

**Significance:** 2
**Argument Clarity:** 2
**Rating:** 3
**Confidence:** 4

**Questions:**

Section 3 details parameter-space symmetries, but do they cause real indistinguishability issues in practice (model storage, versioning, differential privacy attacks)? For instance, although permutation symmetry exists theoretically, do optimizers like Adam break it, pushing trained models toward a preferred “gauge”?

**Alternative Views Section:**

No

**Compliance With Llm Reviewing Policy A Conservative:**

Affirmed.

**Discussion Potential:**

2

**Final Justification:**

The manuscript requires substantial revision and elaboration in multiple sections to enhance clarity. As such, I recommend rejection.

**Paper Summary:**

This position paper addresses the fundamental question of whether machine learning systems possess a stable, identifiable “identity.” The authors argue that current ML practice, including language, tools, and governance frameworks, implicitly treats trained models as stable, identifiable artifacts, yet this practice lacks rigorous philosophical and theoretical grounding.

**Position:**

Yes

**Position In Title:**

Yes

**Related Work:**

2

**Strengths And Weaknesses:**

Strengths:

- The paper challenges a core assumption in ML practice: the ontological status of model identity. This is highly relevant to the ML community, touching on reproducibility, model governance, versioning, and regulatory compliance.

- The paper is well-structured, with arguments developed systematically across functional and structural dimensions.

Weaknesses:

- The paper uses “identity,” “objecthood,” and “individuation” interchangeably without sharp distinctions.

- Section 3 focuses on permutation and continuous symmetries in deep neural networks but undercovers other modern architectures.

- While a position paper does not require extensive experiments, 1–2 concrete examples would strengthen persuasiveness.

**Support:**

2

---

> ### Author Rebuttal · Authors · 2026-03-26
>
> We thank the reviewer for the careful reading and constructive feedback.
>
> **On terminology (“identity”, “objecthood”, “individuation”).**
>
> We agree that these terms are used in a compressed manner. Our intended distinction is:
>
> Individuation: the criteria under which two systems are considered the same or different.
> Identity: the resulting equivalence class under those criteria.
> Objecthood: the assumption that such an equivalence class is stable and context-independent.
>
> The paper’s claim is that modern ML systems do not admit context-independent individuation, and therefore objecthood must be treated as a convention. We will revise the text to make this separation explicit.
>
> **On coverage of architectures.**
>
> Section 3 uses permutation and continuous symmetries as representative mechanisms rather than an exhaustive catalogue. The argument does not depend on specific architectures; it relies on the general property that modern models admit many-to-one mappings from parameters to functions. We will clarify this scope explicitly.
>
> **On the need for examples.**
>
> We agree that anchoring examples would improve clarity. We will incorporate illustrative cases such as (i) permutation-equivalent networks realizing identical functions, and (ii) embedding rotations that preserve outputs under inner-product–preserving transformations. These are structural features of widely used architectures rather than edge cases.
>
> **On the practical relevance of symmetries.**
>
> While optimizers may bias training toward particular regions of parameter space, they do not eliminate the underlying equivalence classes. Different initializations or training trajectories can yield functionally similar systems with distinct internal representations. Our claim concerns identifiability in principle, not whether a specific optimizer selects one representative.
>
> **On empirical grounding.**
>
> In practice, model identity is often defined at the artifact level (e.g., SHA-256 hashes), ensuring bit-level immutability and referential integrity. This notion is operationally robust but strictly finer than functional or structural identity: two artifacts with different hashes may implement the same function or fall within the same symmetry class, while identical hashes guarantee only byte-level equality. When identity is invoked across contexts (e.g., retraining or compression), the relevant equivalence relation should therefore be made explicit.
>
> **On the distinction between “update” vs “new model”.**
>
> A related issue is the distinction between an “update” and a “new” model. In practice this is handled operationally (e.g., versioning), but without explicit criteria. In our terms, this reflects a hierarchy of equivalence relations: a system may be the same under a coarse behavioral tolerance yet different under a finer structural criterion. For instance, quantization may preserve task performance (within tolerance limits) while altering parameters, thereby supporting different identity judgments depending on context. Our claim is that such distinctions depend on the chosen equivalence and should be stated explicitly.
>
> We will revise the manuscript to sharpen terminology and include concrete examples clarifying the argument.

---

> > ### Author Rebuttal · Reviewer_JeLy · 2026-04-02
> >
> > The authors have addressed some of my concerns. However, the specific examples were not explained clearly and lacked detail.

---

### Official Review · Reviewer_f5Zf · 2026-03-13

**Significance:** 3
**Argument Clarity:** 3
**Rating:** 4
**Confidence:** 3

**Questions:**

- (Q1) Regarding Section 3.3 (Normalization): How should identity be defined in cases of intentional model transformations like quantization or pruning? In these cases, parameters are altered by design, yet the artifact is often intended to represent the "same" model optimized for efficiency.

- (Q2) The authors advocate for making equivalence relations explicit as a "minimal obligation". In the authors' view, what constitutes a "sufficient" gauge fix for a complex architecture like a Transformer? Is a partial or heuristic alignment (e.g., canonicalizing attention head order) enough to satisfy governance requirements, or must the fix be mathematically complete?

- (Q3) If functional identity is underdetermined and behavior can only falsify equivalence , does this imply that current safety certifications based on "performance similarity" are fundamentally unreliable for high-risk deployments?

- (Q4) What about model statistical specification proposed in learnware paradigm? (like Learnware of Language Models: Specialized Small Language Models Can Do Big; Learnware: Small Models Do Big)

**Alternative Views Section:**

Yes

**Compliance With Llm Reviewing Policy A Conservative:**

Affirmed.

**Discussion Potential:**

4

**Paper Summary:**

This position paper argues that treating a trained machine learning system as a stable, identifiable "object" is a social and technical convention rather than an inherent property of the system. While current engineering practices and regulatory frameworks (like the EU AI Act) presuppose model identity for traceability and governance, the authors demonstrate that this identity lacks a firm foundation in both functional and structural terms:

- Functional Perspective: Because models are trained on finite data, behavioral equivalence is epistemically underdetermined; agreement on a test set does not guarantee agreement across the entire input space.
- Structural Perspective: Modern architectures possess pervasive redundancies and symmetries (both discrete permutations and continuous rotations), meaning infinitely many distinct parameter configurations can represent the same function.

The paper does not suggest abandoning the concept of model identity but advocates for making the underlying equivalence relations explicit. The authors call for researchers, tool-builders, and regulators to move away from assuming identity as self-evident. Instead, they must define the specific functional tolerances, "gauge-fixing" conventions (canonical coordinate systems in parameter space), or procedural guarantees that justify treating different realizations as "the same" for a given purpose.

**Position:**

Yes

**Position In Title:**

Yes

**Related Work:**

3

**Strengths And Weaknesses:**

- Strengths: This paper addresses the critical issue of defining and tracking AI systems for governance and reproducibility. The authors provide a well-reasoned analysis of how behavioral underdetermination and architectural symmetries make stable model identity technically elusive. This inquiry is highly significant given new regulatory demands, such as the EU AI Act. Rather than being purely critical, the work offers a constructive path forward by advocating for the explicit statement of "equivalence relations" to improve standards in model tracking and evaluation.

- Weaknesses:
  - (W1) Practicality and Scalability of Proposed Obligations: The paper proposes a "gauge fix" as a minimal obligation for structural identification. However, the manuscript remains largely silent on the scalability of this requirement. In modern deep learning, the number of permutation symmetries grows factorially; for a network with $L$ layers and $N$ neurons per layer, there are $(N!)^L$ equivalent representations . While a position paper is not required to provide a technical tool, the lack of discussion regarding whether such a fix is even theoretically or computationally attainable for frontier models (e.g., large-scale Transformers) weakens the practical impact of the proposed position. Without addressing these hurdles, the call to action may be perceived as an idealistic requirement rather than an actionable engineering standard.
  - (W2) Disconnect between Epistemic and Artifact Identity: Current engineering defines identity through bit-level immutability (e.g., SHA-256 hashes of weight files) to ensure referential integrity in deployment pipelines . While the authors correctly argue that this does not capture the model's "knowledge" or epistemic state, the paper should more clearly explain why this existing, highly stable "artifact identity" is insufficient for the legal traceability and accountability requirements discussed in regulatory contexts.
  - (W3) Complexity of Functional Partitioning: The proposal to define "functional bands or partitions" to secure identity risks introducing a new layer of subjectivity. In high-dimensional generative tasks, the criteria for choosing these regions and tolerances may themselves be underdetermined or arbitrary. This potentially shifts the problem of non-identifiability from the model to the "region definition" rather than resolving the underlying ambiguity.

**Support:**

3

---

> ### Author Rebuttal · Authors · 2026-03-26
>
> We thank the reviewer for the thoughtful and constructive assessment, and for recognizing both the relevance of the question and the intent to articulate a possible constructive path.
>
> **(W1) Practicality and scalability of “gauge fixing”.**
>
> This is a fair objection. Our intent is not to require full symmetry resolution, which is generally intractable for modern architectures. The role of “gauge fixing” in the paper is narrower: it makes explicit the equivalence relation under which structural identity is claimed. In many cases, partial or heuristic conventions (e.g., alignment of components or normalization choices) are sufficient for a given application, even if they do not eliminate all symmetries.
>
> Our motivation is informed in part by deployment experience in regulated environments, where issues of model identity already arise and are handled through implicit, domain-specific conventions. The aim of the paper is to surface these conventions and make them explicit, rather than to require a complete canonicalization. We will revise the manuscript to clarify this scope.
>
> **(W2) Artifact identity vs. epistemic identity.**
>
> Hash-based identity plays an essential role in ensuring the integrity of deployed artifacts and supporting traceability. Our claim is that it operates at a different level than the identities implicitly invoked when reasoning about model behavior, retraining, or equivalence across deployments.
>
> In particular, artifact identity is strictly finer than functional or structural identity: two artifacts with different hashes may implement the same function or belong to the same equivalence class under parameter symmetries, while identical hashes guarantee only byte-level equality. If identity claims are transferred across contexts without specifying the underlying equivalence, this distinction becomes consequential.
>
> **(W3) Functional partitioning and subjectivity.**
>
> This is an important observation. The proposal does not eliminate subjectivity; it suggests to make it explicit when functional equivalence is used as the basis for identity. In lower-dimensional or tightly constrained action spaces (e.g., control systems with defined tolerances), partitions can be specified directly. In higher-dimensional settings, equivalence may instead be defined through tolerances in the representation space. In both cases, the tolerance is not intrinsic to the model but part of the chosen equivalence relation.
>
> **(Q1) Identity under transformations (quantization, pruning).**
>
> Whether a transformed model is considered the “same” depends on the chosen equivalence relation. If identity is grounded in behavior and outputs remain within specified tolerances, the models may be treated as equivalent. Under structural criteria, they may not. This reflects a hierarchy of equivalence relations rather than a single notion of identity.
>
> **(Q2) What constitutes a “sufficient” gauge fix?**
>
> A sufficient gauge fix is one that makes explicit the equivalence class relevant to the application. It need not produce a unique canonical representation. For complex architectures, partial or heuristic alignments (e.g., canonical ordering of components) may be sufficient for governance or comparison purposes.
>
> **(Q3) Implications for safety certification.**
>
> Our argument does not imply that current certification practices are invalid. It highlights that they rely on implicit equivalence assumptions (e.g., performance similarity over specified regions). These assumptions are often appropriate, but their scope and limitations should be made explicit, particularly in high-risk settings.
>
> **(Q4) Learnware and statistical specification.**
>
> Learnware-style approaches, which pair models with specifications, are a promising direction. In our terms, such specifications make parts of the equivalence relation explicit (e.g., task scope, expected behavior). Our argument is complementary: even with such specifications, the criteria under which two model instances are considered the same remain context-dependent and should be stated explicitly.
>
> We will revise the manuscript to clarify these distinctions and to better separate artifact-level identity from the context-dependent equivalence relations underlying functional, structural, and process-based notions of sameness.

---

### Official Review · Reviewer_HY1P · 2026-03-24

**Significance:** 1
**Argument Clarity:** 1
**Rating:** 1
**Confidence:** 5

**Questions:**

N/A. I am very unlikely to change my opinion.

**Alternative Views Section:**

Yes

**Compliance With Llm Reviewing Policy A Conservative:**

Affirmed.

**Discussion Potential:**

1

**Paper Summary:**

This paper states that current views of the result of a machine learning process as an object identified by e.g. it's (i) behavior, (ii) process of training, or (iii) internal structure should be made explicit. Whenever one writes about a machine learning model as having having an identity, one should be explicit about which criterion is being used to define that identity.

**Position:**

Yes

**Position In Title:**

Yes

**Related Work:**

1

**Strengths And Weaknesses:**

The stated position is hard to follow, and seems of low relevance to the ICML community since it is not a falsifiable claim as outlined in the call for position papers. I think that it is unlikely to inspire significant discussion in our community, and could be better suited for a philosophical venue.

**Support:**

1

---

> ### Author Rebuttal · Authors · 2026-03-26
>
> We thank the reviewer for the feedback.
>
> **On “non-falsifiability”.**
>
> The paper does not propose a metaphysical claim about identity, but a claim about current ML practice: namely, that treating learned systems as stable objects presupposes an equivalence relation that is rarely made explicit. This claim is falsifiable in a straightforward way: it would be refuted by demonstrating that functional behavior, internal structure, or training process provides context-independent criteria of identity for modern ML systems. Sections 2 and 3 argue that this is not the case, due to behavioral underdetermination and structural non-identifiability.
>
> **On relevance to the ICML community.**
>
> The paper addresses assumptions that are already embedded in core ML workflows, including model versioning, reproducibility, benchmarking, and deployment. These workflows rely on implicit notions of when two models are “the same” or “different” (e.g., across retraining, compression, or evaluation), yet the criteria for these judgments are rarely specified. The contribution of the paper is to make these assumptions explicit and to articulate minimal conditions under which such identity claims can be interpreted.
>
> **On the discussion potential.**
>
> Our intent is not to shift the conversation to a philosophical domain, but to clarify a technical ambiguity that already affects engineering practice and governance, particularly in regulated and high-stakes domains. The distinction between artifact identity (e.g., hashes), functional equivalence, and structural equivalence is routinely encountered in practice, particularly when models are modified, retrained, or audited. Making these distinctions explicit provides a common language for reasoning about such cases.
>
> We will revise the manuscript to clarify the scope of the claim and to make its practical implications more explicit.

---

> > ### Author Rebuttal · Reviewer_HY1P · 2026-04-03
> >
> > I think that the position needs to be sharpened with clear falsifiable claims. I stick by my original review.

---

### Decision · Program_Chairs · 2026-04-30

**Decision:**

Accept (regular)

**Comment:**

I find the overall claim of interest, and not something always trivially understood within the community. While reviewers raise various questions about the work, the authors rebuttal in my view answer most of these concerns. I believe the position is not of interest to the community and given new insights with regard to defining regulation but also to understand and compare models.

Given the strong argumentation, clarity of the write-up noticed by multiple reviewers, and importance of the position, I think this work can be quite valuable to the community.